# Structure and Mechanical Behavior of Heat-Resistant Steel Manufactured by Multilayer Arc Deposition

**Ilya V. Vlasov \***, **Antonina I. Gordienko, Aleksandr V. Eremin**, **Vyacheslav M. Semenchuk** and **Anastasia E. Kuznetsova**

Institute of Strength Physics and Materials Science, Siberian Branch of Russian Academy of Sciences, 634055 Tomsk, Russia; mirantil@ispms.ru (A.I.G.); ave@ispms.ru (A.V.E.)

**\*** Correspondence: viv@ispms.ru

**Abstract:** The manuscript demonstrates the structure and the mechanical behavior of a material manufactured by multilayer arc deposition. Three-dimensional printing was performed using OK Autrod 13.14 wire on a substrate of heat-resistant 12Cr1MoV steel in the standard gas metal arc welding (GMAW) mode and in the coldArc mode with reduced heat input. The printed materials have 40–45% higher strength and 50–70% lower ductility compared to the substrate. The microhardness of the printed materials is higher than the substrate, but it is reduced at the transition regions between the deposited layers. These regions have been studied using optical microscopy and digital image correlation. Such layer boundaries are an additional factor in reducing the plasticity of the material. The increase in strength and decrease in ductility for printed materials compared to the ferrite–pearlitic substrate is associated with a high cooling rate and the formation of a mixture of acicular and allotriomorphic ferrite, which have higher hardness. The structure of the obtained layers along the height is non-uniform and undergoes changes during the deposition of new layers. The main difference between the 3D printing modes is the reduced heat input in the coldArc mode, which results in less heat accumulation and faster cooling of the wall. Thus, a more dispersed and solid structure was formed compared with GMAW. It was concluded that the cooling rate and the level of heat input are the main factors affecting the structure formation (martensitic, bainitic, or ferritic), the height and quality of the surface, and the mechanical properties of the printed wall.

**Keywords:** wire arc additive manufacturing; additive manufacturing; heat-resistant steel; microstructure; mechanical properties; heat input

## 1. Introduction

Additive manufacturing (AM) has made it possible to develop new approaches to the design of parts, their production, and the combination of different materials within a single component [1,2]. However, a number of problems revealed were related to the porosity of products, defects (microcracks, discontinuities), and formation of inhomogeneous structure associated with the cyclic heating during the manufacturing process [3]. In comparison to the conventional manufacturing of metal products using casting and forging, additive manufacturing is quite new and is actively developing for the involvement in the production process. The possibilities of AM are economically feasible; it saves approximately 50% of the cost of manufacturing compared to traditional production methods [4,5]. For manufacturing of low-volume, unique, high-tech, or complex shape components, AM is indispensable and allows the use of new designs with optimized topology.

The application of molten metal is usually carried out either in a vacuum or an inert gas environment that protects the material from oxidation. The vacuum makes it possible to effectively protect materials sensitive to embrittlement by eliminating interaction with hydrogen, oxygen, and nitrogen. However, a more efficient approach is a deposition using Metal Inert Gas (MIG) or Metal Active Gas (MAG) techniques. It is suitable for

most materials and applications, including 3D printing of difficult-to-machine alloys, such as titanium [6], low-alloy structural steels [2,7], austenitic [8,9] or martensitic steels [10], aluminum [11–13], and bronze [14–16].

There are several powder-based approaches for AM, such as selective laser melting, laser-directed energy deposition, and plasma arc additive manufacturing, which have been actively developed in recent years.

The widespread, high-performance, economical, and easy-to-manufacture technology is Wire Arc Additive Manufacturing (WAAM) [2]. It makes it possible to achieve a material deposition rate of the order of 4–9 kg/h [17] and to produce large-scale structures. This 3D printing technique significantly saves expensive raw materials and reduces the cost of milling and refining printed components. WAAM utilizes standard welding equipment, while the welding modes are modified to achieve optimal performance for layers deposition.

WAAM technology is divided into Gas Metal Arc Welding (GMAW), Gas Tungsten Arc Welding (GTAW) and Plasma Arc Welding (PAW). The last two methods allow printing with two or more feeding wires, thus obtaining unique alloys or combining layers to produce composite materials (combined wire–powder printing) [18].

Electron beam additive manufacturing and wire arc additive manufacturing in a shielding gas environment (Gas Metal Arc Welding—GMAW) are used for the efficient production of components [2]. GMAW is the most common and easy to implement due to fast layering process and absence of production waste (such as used powder), and it does not require high vacuum as electron beam additive manufacturing.

In a number of papers, it was shown that products from low-carbon steels obtained using GMAW have nearly the same mechanical properties as forged steel [19,20]. This indicates the promise of this technology and the need for further optimization of technological parameters for industrial application.

It is noted that it is often possible to distinguish the formation of different zones and structural inhomogeneities when forming products by the GMAW method. For example, structure inhomogeneity for SS316L steel along the wall height was found, and three zones were identified that differ in microstructure [20–22].

In order to control the structure and properties of products during 3D printing by the GMAW method, a number of investigations were carried out to study the influence of welding torch travel velocity. M. Dinovitzer et al. [23] found that an increase in the travel velocity and a decrease in the arc current lead to a decrease in the melting depth, thus changing the microstructure of the material. Xiong et al. [24] showed that the surface quality of a low-alloy steel wall can be improved by increasing the torch travel velocity. M. M. Tawfik et al. [25] obtained a more dispersed and equiaxed microstructure of the specimen made from the Al–Mg alloy by the GTAW–WAAM method. This is due to the high torch travel velocity, which is directly related to the level of heat input.

Thus, heat input plays an important role in the formation of the structure and mechanical properties of components printed by GMAW. It can be controlled through the torch movement to obtain a defect-free layer (avoidance of pores, lack of penetration, cracks, etc. [26,27], and high internal stresses [7]), while the rest of the 3D printing parameters are related to the welding process. To obtain a defect-free material, "welding" parameters (current intensity, voltage, and wire feed rate) are selected, which are synergistically related and used for a specific welding process. They do not take into account the processes of multiple heat input and structural transformations as a result of heat accumulation. Therefore, the task of obtaining a defect-free material and reducing uncontrolled thermal effects is not an easy one.

Another way to control the heat input for GMAW is printing modes such as Cold Metal Transfer (CMT) (Fronius) and coldArc (EWM) [28]. The principle of these technologies is to weld with a short electrical arc as well as changing cycles of short circuit and arc burning. In CMT, the additional reciprocating motion of the wire takes place, coordinated with the short circuit. Modes of reduced heat input were developed for welding of thin sheets in order to

avoid their burn-through. These regimes turned out to be well applicable for printing bulk products, where deep metal melting and a large weld pool are not necessary [29–31].

The prospects of using CMT wire printing technology from low-carbon structural steels are shown in the literature [32–34]. CMT products have nearly the same mechanical properties as specimens made by conventional techniques, such as rolling, forging, etc. It is possible to save the impact strength and to achieve a more uniform distribution of hardness in different sections of the printed product. V. T. Le et al. [35] showed that walls fabricated with low heat input have better mechanical properties and less surface roughness. B. P. Nagasai et al. [36] showed that the products obtained using the CMT process have a finer grain and higher mechanical characteristics than products obtained using the standard GMAW.

However, the GMAW technology has limitations. Due to the non-uniform melting of the wire and partial spreading of the layer, a high surface roughness occurs in wire-feed methods. Another disadvantage is the cyclic thermal effect of new layers on previous ones, which affects the structure and mechanical properties of the product.

The problems that arise might be solved by investigation of thermal exposure and structural transformations in the 3D printing [2,8], or by developing methods to optimize printing with a short arc and reducing heat input [37]. The other way is thermal treatment to improve the structure after 3D printing [38]. In the observed manuscripts, standard GMAW and heat input modes (mainly CMT) were compared. However, the literature analysis shows that not enough attention is paid to the processes occurring in the structure of the product during multilayer deposition. Thermal cycling and changing conditions for heat removal along the product height lead to variety in structure and properties in one product [19,39]. Different levels of heat input have a significant effect on the cooling rate of the layer and the conditions for the formation of the microstructure.

Thus, it is important to predict the distribution of heat in the product and adjust the printing modes to control the process of structure and phase formation. A detailed analysis of structure evolution during printing is necessary to design manufacturing strategy to achieve required mechanical properties and optimal structure.

The structure of high-performance alloys is selected based on the operating conditions. It includes a large number of alloying elements and multi-stage heat treatment. One of these alloys is heat-resistant steel, which can be divided into martensitic, austenitic, and pearlitic. They are also used to manufacture components of steam pipelines and heating and cooling systems, such as vapor coolers, separators for steam and compressed air, superheaters, etc. Most designs require a large contact area for efficient heat transfer. The complex geometry of components obtained using additive manufacturing excludes inhomogeneity, e.g., joints. Joints play a role in macro inhomogeneities where the fracture origins. Thus, it will be possible to significantly reduce production costs, enhance structural efficiency, and extend the service life.

Most of the work is focused on austenitic alloys with corrosion resistance properties [2,20,40–46], whereas heat-resistant pearlitic steel 12Cr1MoV attracts less attention and insufficient number of works devoted to the additive manufacturing of products from heat-resistant pearlitic steel.

Heat-resistant pearlitic steel 12Cr1MoV is widely used for manufacturing superheater pipes, forgings of steam boilers and pipelines, components of gas turbine cylinders, etc. The alloying elements of the steel are vanadium, chromium, and molybdenum. The last two provide strength through a substitutional solution with iron and form carbides such as $M_3C$. Vanadium results in the dispersion strengthening of steel due to the formation of VC. Thus, the thermal stability of the steel is achieved during dispersion hardening due to the precipitation of carbides under heat treatment. The precipitated dispersed particles in the ferrite grains hinder creep processes at high temperatures [47–49].

Welding wire "OK Autrod 13.14" was used for surfacing and welding structural elements made of 12Cr1MoV steel. This wire is recommended [50] for welding heat-resistant pearlitic steels and, in particular, for the 12Cr1MoV steel being investigated in

this paper. It includes the main alloying elements of 12Cr1MoV steel with the addition of manganese and silicon to optimize the structure during the welding process. "OK Autrod 13.14" is slightly inferior to austenitic steels in terms of maximum operating temperature while being much cheaper and can be used as an analogue in cases where corrosion resistance properties are not required. Thus, 3D printing of a wall in this work has been performed using OK Autrod 13.14 wire.

The aim of the present manuscript is to study the structure, mechanical properties, and deformation behavior of specimens cut from the wall printed by the multilayer arc deposition method under various heat input modes using pearlitic steel wire. The paper will evaluate the effect of heat input in various 3D printing modes (standard GMAW and coldArc mode with reduced heat input). The research will be carried out on the conditions for the formation of the structure and mechanical properties, as well as an assessment of the deformation behavior of the printed material and the influence of layer boundaries on mechanical properties.

## 2. Materials and Methods

Copper-plated welding wire "OK Autrod 13.14" (ESAB Corporation, North Bethesda, MA, USA) with a diameter of 1.2 mm was used for multilayer deposition. Copper is used as a protective coating of the wire and is not included in its composition, as indicated in the handbooks. During welding, copper quickly evaporates, and less than a percent remains in the hardened material. A 10 mm plate made of 12Cr1MoV steel was a substrate. It has similar chemical composition to the wire and is used to manufacture components operating at temperatures of 540–580 °C. The substrate steel was also a reference material to compare the mechanical properties of the printed walls. The chemical compositions of substrate and wire OK Autrod 13.14 were received from the supplier and were based on the inspection certificate, which guarantees the quality of the materials (Table 1).

**Table 1.** Chemical composition of the 12Cr1MoV steel substrate and OK Autrod 13.14 wire (%).

| Material | C | Si | V | Cr | Mn | Fe | Ni | Cu | Mo | P/S/N |
|---|---|---|---|---|---|---|---|---|---|---|
| 12Cr1MoV (GOST 5520-79) | 0.12 | 0.23 | 0.169 | 0.98 | 0.47 | bal. | 0.18 | – | 0.261 | <0.014 |
| OK Autrod 13.14 (GOST 2246-70) | 0.06–0.1 | 0.45–0.7 | 0.2–0.35 | 0.95–1.25 | 1.2–1.5 | bal. | <0.3 | – | 0.5–0.7 | <0.025 |

Three-dimensional printing was carried out with metal wire via the electric arc deposition shielded by a mixture of 82% Ar and 18% $CO_2$. The apparatus consists of a FANUC AM-100iD multi-axis mechanized manipulator (FANUC Corporation, Oshino-mura, Japan) equipped with an R-30iB Plus controller and a Roboguide WeldPro software (FANUC Corporation, Oshino-mura, Japan). The manipulator uses an EWM Titan XQ R 400 welding machine (EWM GmbH, Mündersbach, Germany) with wire feeder M drive 4 Rob 5 HW XH LI (EWM GmbH, Mündersbach, Germany) designed for robotic welding (Figure 1). The layers were applied in two modes in the standard mode according to the GMAW process and in the coldArc mode, which has a reduced heat input. The set of synergistic curves provided by the manufacturer was analyzed for selection of the optimal printing parameters to ensure stable arc burning, minimal spatter, and a homogeneous deposited layer.

Multilayered arc deposition was carried out from the wall with a length of 100 mm and consisted of 33 layers (Figure 2a,c,d). The inclination angle of the welding torch relative to the substrate was 10° (the direction of movement was a "backward angle"). The layers were deposited with an offset of 1 mm to increase the wall thickness. The parameters of the modes used and the resulting wall dimensions are presented in Table 2. The heat input (*HI*) was evaluated in kJ/mm by the equation from [29]:

$$HI = \eta \cdot V \cdot I \cdot 60 \ / \ (S \cdot 1000) \tag{1}$$

where *S* is the travel velocity of the torch, mm/min; *I*—average current intensity, A; *V*—arc voltage, V; *η*—efficiency, which in this case is equal to 0.8.

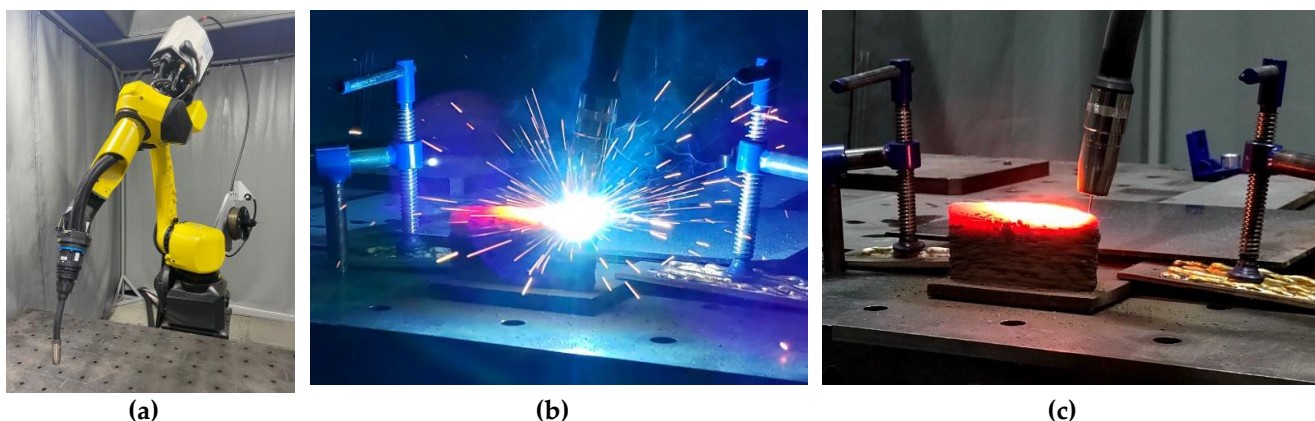

**Figure 1.** Images of printing equipment FANUC AM-100iD with a welding torch (**a**), deposition process (**b**), and cooling of the layer after deposition (**c**).

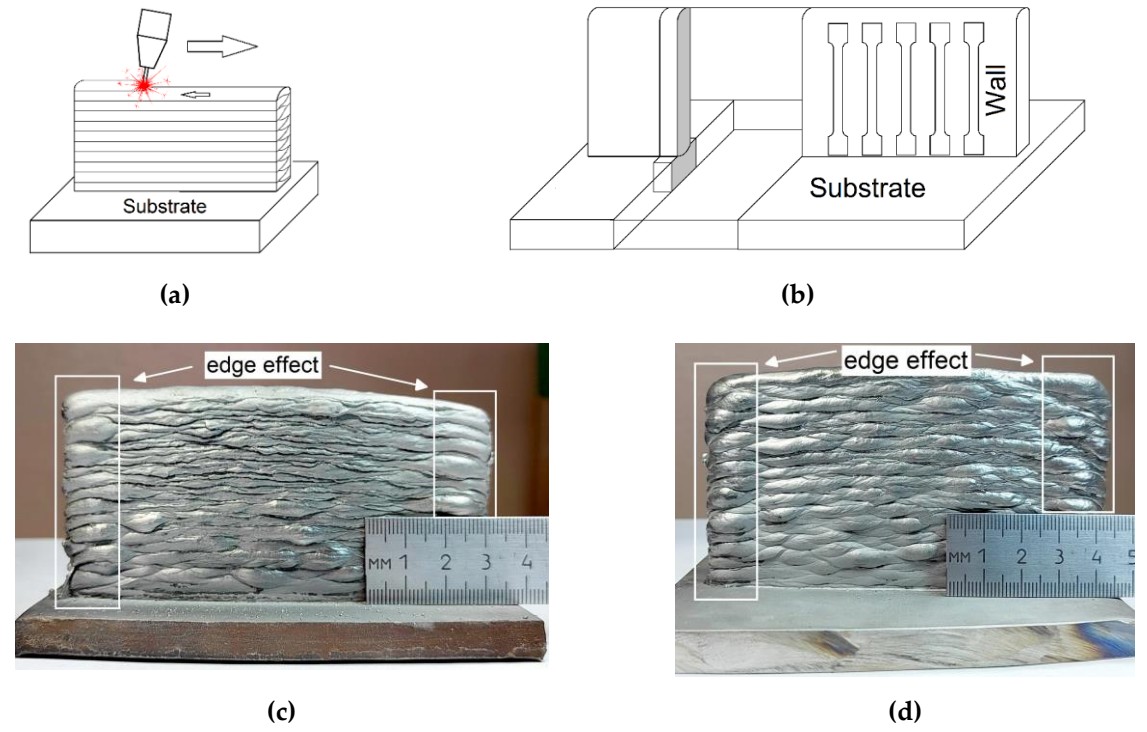

**Figure 2.** Printing scheme of the wall and the layout for the samples cut out (**a**,**b**); walls obtained using GMAW (**c**) and coldArc (**d**) technologies.

Specimens for mechanical and structural tests, as well as microhardness measurements, were cut out of the wall using an electrical discharge machine, as shown in the diagram in Figure 2b. For mechanical tensile tests, the specimens had the dog-bone shape with the dimensions of the reduced section of 4 mm × 1.5 mm × 40 mm (Width × Thickness × Length). Electrical discharge machining provides high precision of specimen dimensions and low surface roughness. However, the post-cut polishing was performed in several steps—emery paper with a P240 grade; P600; P1000; P2000, and finally, it was polished by diamond paste. To analyze the microstructure and measure the microhardness, a transverse section of the wall and substrate was cut out.

**Table 2.** Three-dimensional printing conditions and wall dimensions manufactured at two modes.

| Parameters | GMAW | coldArc |
|---|---|---|
| Current intensity, A | 135 | 128 |
| Voltage, V | 16.4 | 15.8 |
| Wire feed rate, mm/min | 3500 | 3500 |
| Torch travel velocity, mm/min | 250 | 250 |
| Gas feed rate, l/min | 15 | 15 |
| Dwell time between layer deposition, sec | 40 | 40 |
| Heat input (HI), kJ/mm | 0.425 | 0.388 |
| Wall thickness, mm | $8 \pm 1$ | $7 \pm 1$ |
| Wall height, mm | 53 | 58 |

Static tensile tests were carried out on an Instron 5582 electromechanical machine (Illinois Tool Works Inc., Glenview, IL, USA) with a crosshead travel speed of 1.5 mm/min. Three to six specimens for each type of material were tested. A digital image correlation method was used to obtain detailed information about the mechanical behavior of the material and also plays a role of an optical extensometer. A speckle pattern consisting of a white background and black randomly located dots was applied to the surface of the sample. The specimens were photographed at a rate of 1 frame per second during the tensile tests. Based on the obtained photographs, the deformation fields were calculated using the VIC 2D software ( Correlated Solutions Inc., Irmo, SC, USA). As far as the material was printed in layers to illustrate the behavior of the layers, the analysis of the field along the line "AB" in the center of the specimen was used (Figure 3a). The longitudinal strain component $\varepsilon_{yy}$ along the line was displayed as a graph (Figure 3b).

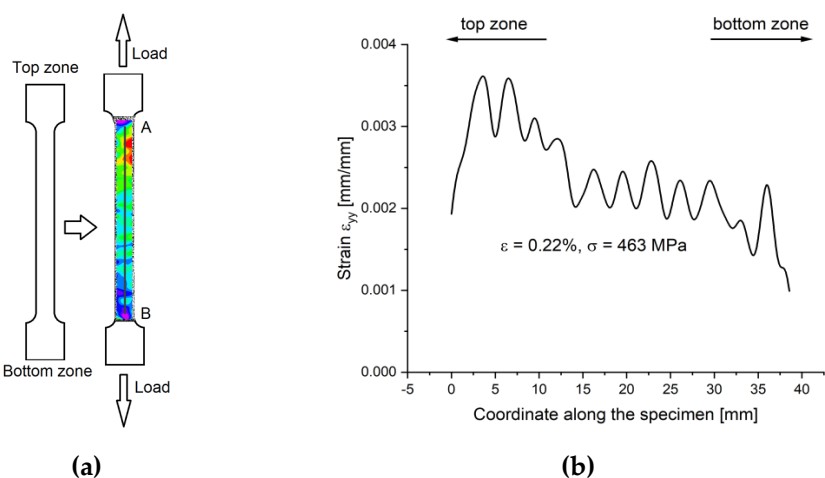

(a)                                    (b)

**Figure 3.** Scheme of data extraction from strain fields of the sample: (**a**) Strain field obtained by DIC with the central line for data extraction; (**b**) graph of the $\varepsilon_{yy}$ stain component extracted from the central line of the strain field.

The structure of the specimens was studied using a Carl Zeiss Axiovert 25 microscope (Carl Zeiss AG, Oberkochen, Baden-Württemberg, Germany) and a LEO EVO 50 scanning electron microscope (Carl Zeiss AG, Oberkochen, Baden-Württemberg, Germany) at the Center for Collective Use "NANOTECH" of the ISPMS SB RAS. Samples were etched with a 5% nitric acid solution. Microhardness was evaluated on a PTM-3 device with a load on the Vickers pyramid of 0.98 N (100 g). The measurements were performed over the entire height of the cross section, including the substrate, with an average step of 5 mm and at least 5 indents in each zone.

## 3. Results and Discussion

### 3.1. Wall Appearance

The torch travels along the wall during 3D printing (Figure 2a). Non-uniform spreading of the metal occurs, thus leading to a "wave-like" formation of layers (Figure 2c,d). The wave-like nature of the layers formed is associated with the inhomogeneity of the deposition of molten metal. This includes arc instability, arc distortion, and possible micro-damage of the wire, which can lead to slight scatter in the deposition process. The heat remains in the wall after welding, and thus prolongs the time of solidification for the deposited metal. So, the liquid metal could spread unevenly from layer to layer. Local distortions of the geometry and excessive spreading of the layer accumulate during printing. This eventually leads to the formation of undulating layer boundaries. The increase in the cooling time between layers could be a possible solution to this issue. The other is to form a thick wall where adjacent layers would limit the horizontal spread of the new one.

In addition, the torch stops at the left and right edge of the wall dwelling for 40 seconds and then moves in the opposite direction, forming the next layer. This results in increased heating of the lateral parts of the wall and, hence, stronger spreading of the metal at edges. The length of this "edge effect" is estimated at 10–15 mm. In the center of the wall, the layers are formed more evenly.

The average vertical travel of the welding torch after each layer deposition was 1.6 and 1.8 mm for the GMAW and coldArc modes, respectively. However, as the height of the wall increased, vertical travel of the torch had to be adjusted during printing to achieve a constant arc length. Non-uniform spreading of the layer and deposition of layers with a horizontal offset led to waviness and fluctuations in the layer thickness. Thus, the thickness of the layers is not the same along the height of the wall.

Compared to coldArc mode, GMAW has thinner layers. The higher layer height in coldArc mode is caused by less metal spreading due to lower heat input 0.425 kJ/mm in GMAW mode versus 0.388 kJ/mm in coldArc mode. This explains the difference in the geometric dimensions of the walls and contributes to the formation of a higher and narrower wall in the coldArc printing mode (Table 2).

### 3.2. Metallography

The initial steel structure of the substrate consists of ferrite and pearlite (Figure 4a,b) with an average ferrite grain size of $18 \pm 2$ µm. Pearlite has a granular morphology and is located along the boundaries of ferrite grains in the form of interlayers and separate clusters.

Single pores smaller than 1 µm in size are observed in the cross section of the printed walls. They have a regular round shape and were formed as a result of various factors, such as insufficient gas shielding, microscopic wire defects, etc. The authors believe that their small number and size should not have a noticeable effect on the mechanical properties of the samples.

The schematic arrangement of the cross section of the wall during optical studies of the structure is shown in Figure 4c. Figure 4e, h shows the microstructures of the lower part of the walls and the substrate obtained in the GMAW and coldArc modes. A pronounced thermal effect and heat-affected zone on the substrate is observed at a distance of 700–1000 µm from the wall–substrate boundary. At a greater distance, the structure completely transforms into a ferrite–pearlite one.

For both modes of 3D printing, a thin white stripe (Figure 4e,h, indicated by an arrow) 50 µm thick with a dispersed structure of lath martensite is clearly visualized. The formation of the strip is associated with a high-temperature gradient at the wall–substrate interface due to rapid heat removal to the cold substrate and metal table.

One more region is situated in the heat-affected zone (HAZ) of the substrate at a distance of up to 150–200 µm from the wall–substrate boundary. A mixed structure is observed in this region for both GMAW and coldArc. It consists of lath martensite, lath bainite, and areas of granular pearlite. Carbide chains of 0.6–1.7 µm in size are located along the boundaries of lath bainite packets (Figure 4f,i). Despite the nucleation of fragments of

the lath structure, the size of the packets and their boundaries correspond to the size of the original ferrite grains. Therefore, structure inheritance occurs in this area after thermal cycling. The structure in the HAZ changes from martensitic–bainite to ferrite–bainitic with distance from the boundary.

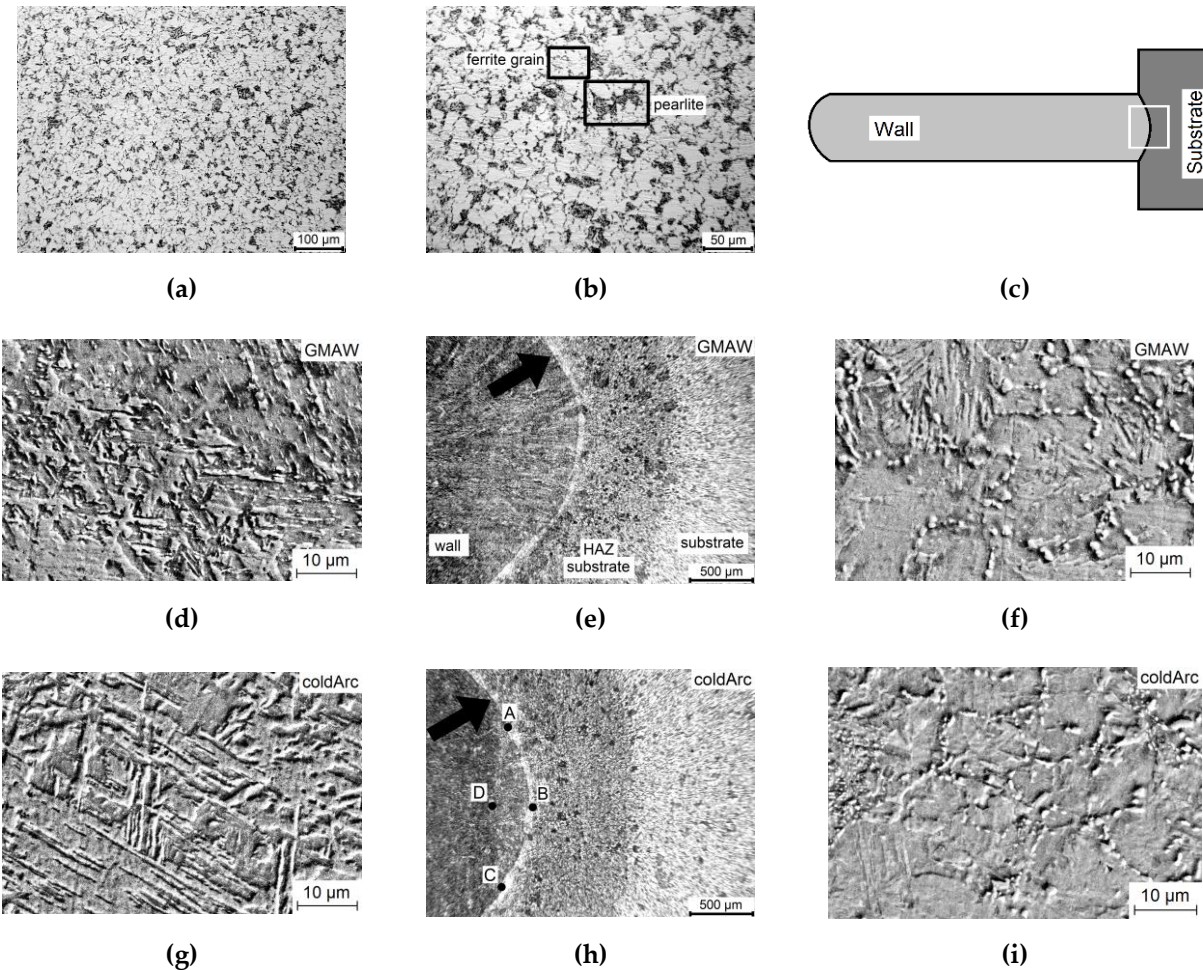

**Figure 4.** Substrate microstructure (**a**,**b**); area on the wall for the investigation (**c**); the lower part of the walls near the wall–substrate boundary (**d**,**g**); wall–substrate boundary (**e**,**h**); HAZ near wall–substrate boundary (**f**,**i**).

The formation of martensitic and transitional structures in the HAZ of the substrate, as well as bainite structures at the lower part of the wall, is associated with uneven heating of these zones and a change in the cooling rate. So, for the first layers, a high heat removal to the substrate significantly increased the cooling rate of this region and ensured the formation of a martensitic structure. However, with the accumulation of heat in the substrate and deposition of new layers, the cooling rate gradually decreased, forming a gradient transition from martensitic to bainitic structure.

A martensitic structure formed in the wall was obtained by the GMAW mode at a distance of up to 100 µm from the boundary (Figure 4d). At larger distance from the substrate (more than 100 µm), the heat removal rate decreases, and as a result, predominantly bainitic structures in the form of acicular ferrite were formed. Columnar grains are visible at the macrolevel in the region close to the fusion boundary. The grains were formed as a result of high crystallization rate in the direction of heat removal (Figure 4e). The boundaries of these elongated grains are the boundaries of the former austenite grains, within which, at the resulting cooling rate, an acicular ferrite structure has formed. In some

cases, allotriomorphic ferrite was isolated at the boundaries of such grains, but it is more common in the middle and upper parts of the walls.

In the case of the coldArc mode, the region with a martensitic structure near the fusion boundary is clearly visible in Figure 4h (region ABCD) and reaches 260 µm between "DB". Elongated boundaries of the former austenitic grains, similar to those observed in the case of the GMAW regime, were not revealed. To the left of the martensitic region ABCD (Figure 4g), the structure in the wall is quite homogeneous and consists of acicular ferrite with a small fraction of allotriomorphic ferrite.

The wall was partially cooled after each printing pass; however, as the number of layers increased, the cooling rate in each new layer decreased due to the heating of the wall, substrate, and table. As a result, a finely dispersed mixture of ferrite grains and acicular ferrite is formed at a distance of 10–15 mm from the substrate for both printing modes (Figure 5a,b). At a distance of 15–30 mm, enlargement of the structural elements with a size of about $80 \pm 10$ µm is observed (Figure 5c–f). Allotriomorphic ferrite was isolated at their boundaries, and acicular ferrite was formed inside the grains. The formation of interlayers of allotriomorphic ferrite is associated with the diffusion processes during austenitic decomposition.

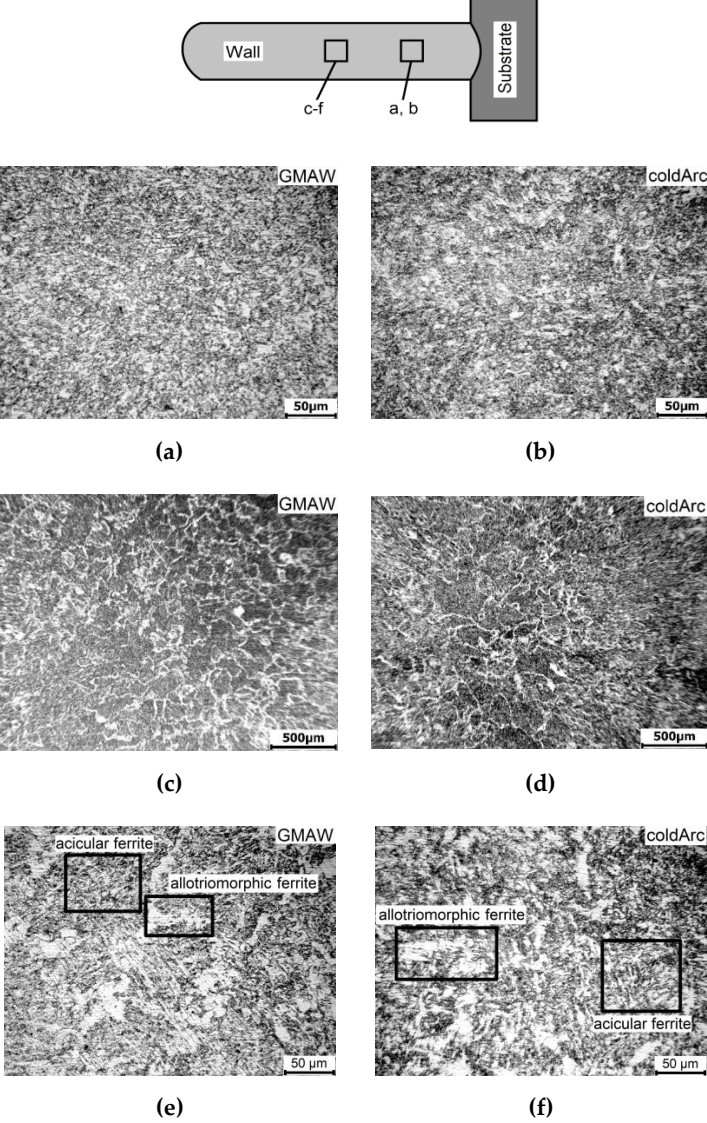

**Figure 5.** Microstructure of the cross section of the middle part of the walls at the distance of 10–15 mm (**a**,**b**) and 15–30 mm (**c**–**f**) from the substrate for GMAW (**a**,**c**,**e**) and coldArc (**b**,**d**,**f**) modes.

The upper part of the wall (0–10 mm from the top) for both printing modes is characterized by the presence of large grains with a size of up to 290 ± 40 μm (Figure 6a,b). They are formed due to heating of the last layer to a high temperature during the printing process. As a result, complete recrystallization occurs, followed by the growth of the austenite grains. However, rapid heat removal to the underlying layers and the atmosphere leads to the formation of an acicular ferrite structure inside the grains (Figure 6c,d). At the very top of the wall, large grains are elongated in the direction of the heat transfer.

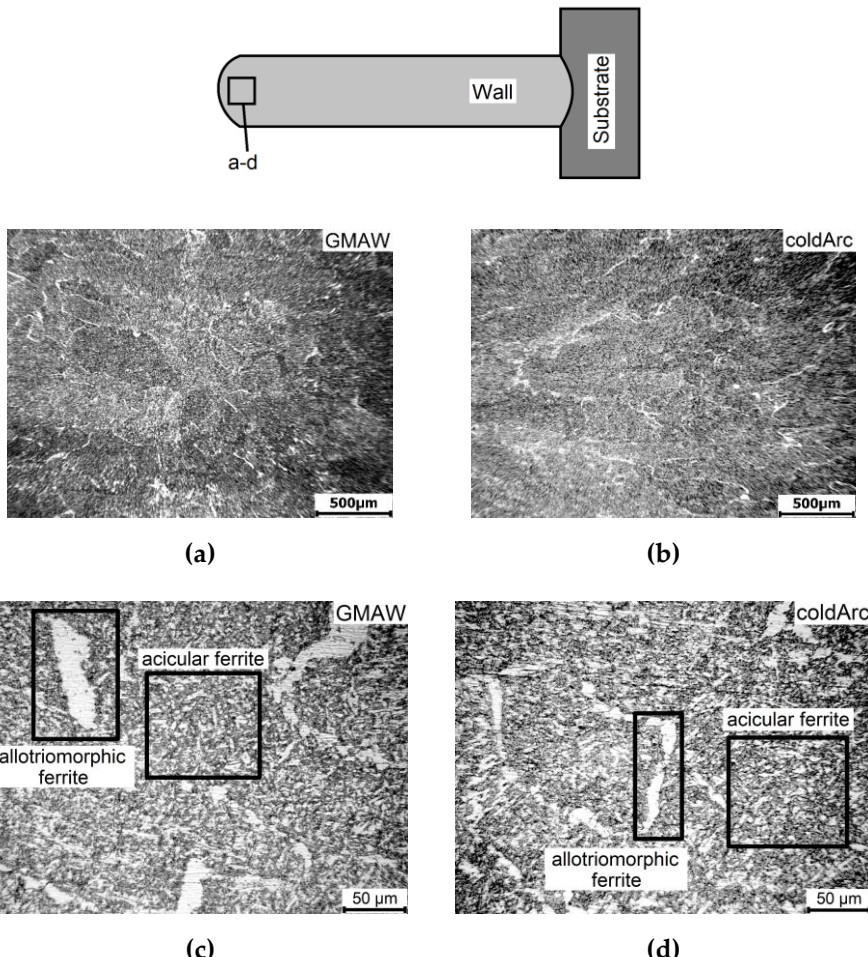

**Figure 6.** Microstructure of the cross section of the upper part of the walls at low (**a**,**b**) and high (**c**,**d**) magnification for GMAW (**a**,**c**) and coldArc (**b**,**d**) modes.

At a distance of more than 10 mm from the top of the wall, the most intensive decrease in grain size occurs (up to 80 ± 10 μm). This difference can be explained that at a depth of more than 10 mm from the top, as a result of thermal cycling, repeated crystallization processes occur. Thus, the grain size decreases.

The presence of macroheterogeneities was found in the structure of the boundaries of the wall layers (Figure 7). These areas contain more of the ferrite phase. The formation of a ferrite phase at the layer boundaries is associated with the maximum heating of these areas during layer printing. Due to this, the cooling rate between the layers decreases, and the formation of ferrite grains occurs. The layer height in the wall after printing in coldArc mode was higher on average, as can be seen from the photographs (Figure 7).

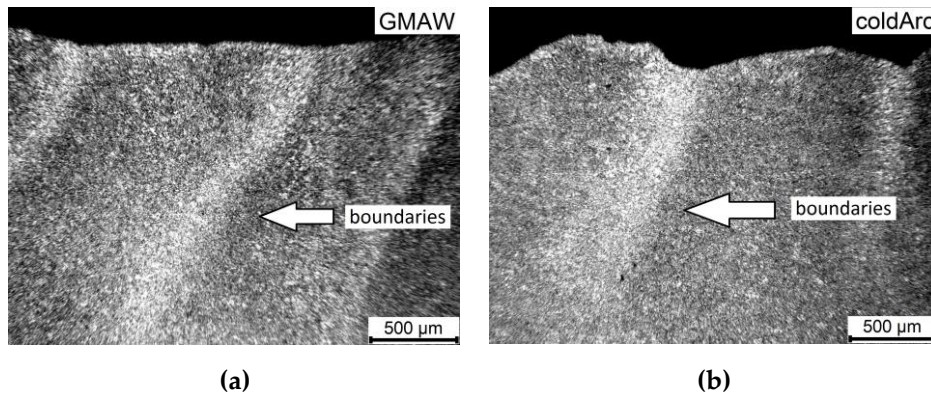

**Figure 7.** Microstructure of the boundaries of the wall layers in the cross section for GMAW mode (**a**), and coldArc mode (**b**).

### 3.3. Microhardness

The microhardness values were measured along the height of the wall and the substrate in cross section (Figure 8); $L = 0$ corresponds to the lower surface of the substrate, which was in contact with the table. A single point with $L < 0$ corresponds to the initial microhardness of the substrate before printing. Measurements of microhardness along the wall height were carried out with a step of 5 mm. However, in the lower part of the wall and HAZ, where microhardness changes significantly, additional indentations were made. The step of measurements in these regions was less than 1 mm. Some of the points in Figure 8 were excluded in order to make the graph clearly visible and not overloaded with excessive data points.

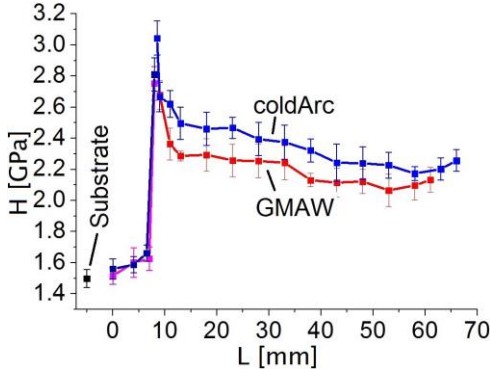

**Figure 8.** Measurement of microhardness along the wall height $L$.

The maximum values of microhardness were found at the substrate–wall boundary, where the highest rate of heat removal is achieved. Such high microhardness is associated with the martensitic structure. Moving from the lower part of the wall, the rate of heat removal gradually decreased as the substrate and table were heated up. As a result, a finely dispersed structure was formed (Figure 5a,b), which is usually characterized by a lower microhardness (Figure 8 from $L\sim10$ to $L\sim25$ mm). Gradually, with an increase in the number of deposited layers and heat accumulation, the cooling rate decreased, which led to an increase in the size and proportion of precipitated ferrite (Figure 5e,f) and a decrease in microhardness (Figure 8 from $L\sim25$ to $L\sim50$ mm).

There is a slight increase in microhardness in the upper part of the wall compared to the central part ($\sim10$ mm, or $L = 50$–70 mm in Figure 8). This is due to the formation of an acicular ferrite structure and a small proportion of ferrite (Figure 6) that did not have time to precipitate as a result of limited thermal cycling (Figure 5e,f).

It should be noted that in the coldArc mode, the microhardness is higher, which is caused by a lower degree of heat input, a higher cooling rate, and the formation of large internal stresses in martensite.

Additionally, microhardness was measured at the layer boundaries (Figure 7). On average, the values at the boundaries were lower than those inside the layers by 15%. This is due to the additional precipitation of ferrite in the boundary regions, which usually has lower strength and hardness.

When comparing the printing modes, it can be seen that in the main part of the wall, the coldArc has higher microhardness than GMAW by about 7%. This is due to less heat input during printing, less heat accumulation in the wall, and faster cooling of the metal. As a result, the proportion of bainite structures becomes larger for coldArc mode, while the proportion of ferrite regions decreases. These differences are not significant, and it is impossible to observe them visually, but they are manifested in microhardness measurements.

### 3.4. Static Tensile Tests

Specimens cut out from the walls demonstrate higher values of ultimate strength and lower strain at break under tension compared to the substrate specimens (Figure 9 and Table 3). First of all, such differences are associated with the rapid cooling of the wall during printing and the formation of dispersed bainitic structures.

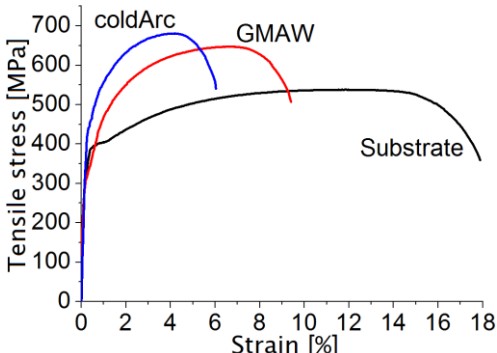

**Figure 9.** Stress–strain diagrams of tested specimens.

**Table 3.** Summary of static tensile tests.

| Material | Yield Point (0.2), MPa | Ultimate Tensile Strength, MPa | Strain at Break, % | Fracture Location from the Fillet, mm | Distance between Localization Peaks, mm |
|---|---|---|---|---|---|
| Russian GOST 5520-79 for 12Cr1MoV | 295 | 440–640 | 21 | - | - |
| Russian GOST 2246-70 for "OK Autrod 13.14" | 600 | 700 | 16 | - | - |
| 12Cr1MoV substrate | 364 ± 19 | 384 ± 21 | 17.8 ± 1.7 | 2.1 | no peaks |
| OK Autrod 13.14 in GMAW mode | 324 ± 18 | 647 ± 39 | 9.3 ± 1.1 | 7 | 4.3 |
| OK Autrod 13.14 in coldArc mode | 460 ± 23 | 681 ± 34 | 5.7 ± 0.8 | 7.2 | 3.2 |

The decrease in plasticity and increase in the yield strength and strength of samples printed in the coldArc mode is obviously associated with a faster cooling rate due to lower heat input compared with the GMAW mode. Similar differences were shown when measuring microhardness, where the wall printed in coldArc mode also showed higher mechanical properties.

Images of the specimen surface were taken and processed to construct strain fields and then analyze inhomogeneities in deformation behavior during tension (Figure 10). Graphs presented in Figure 10a correspond to the strain fields in Figure 10d–f, Figure 10b— Figure 10g–i, and Figure 10c—Figure 10j–l. In such a way, an overall strain field and

associated plot in the central line demonstrate the deformation inhomogeneities at different stages of tensile loading.

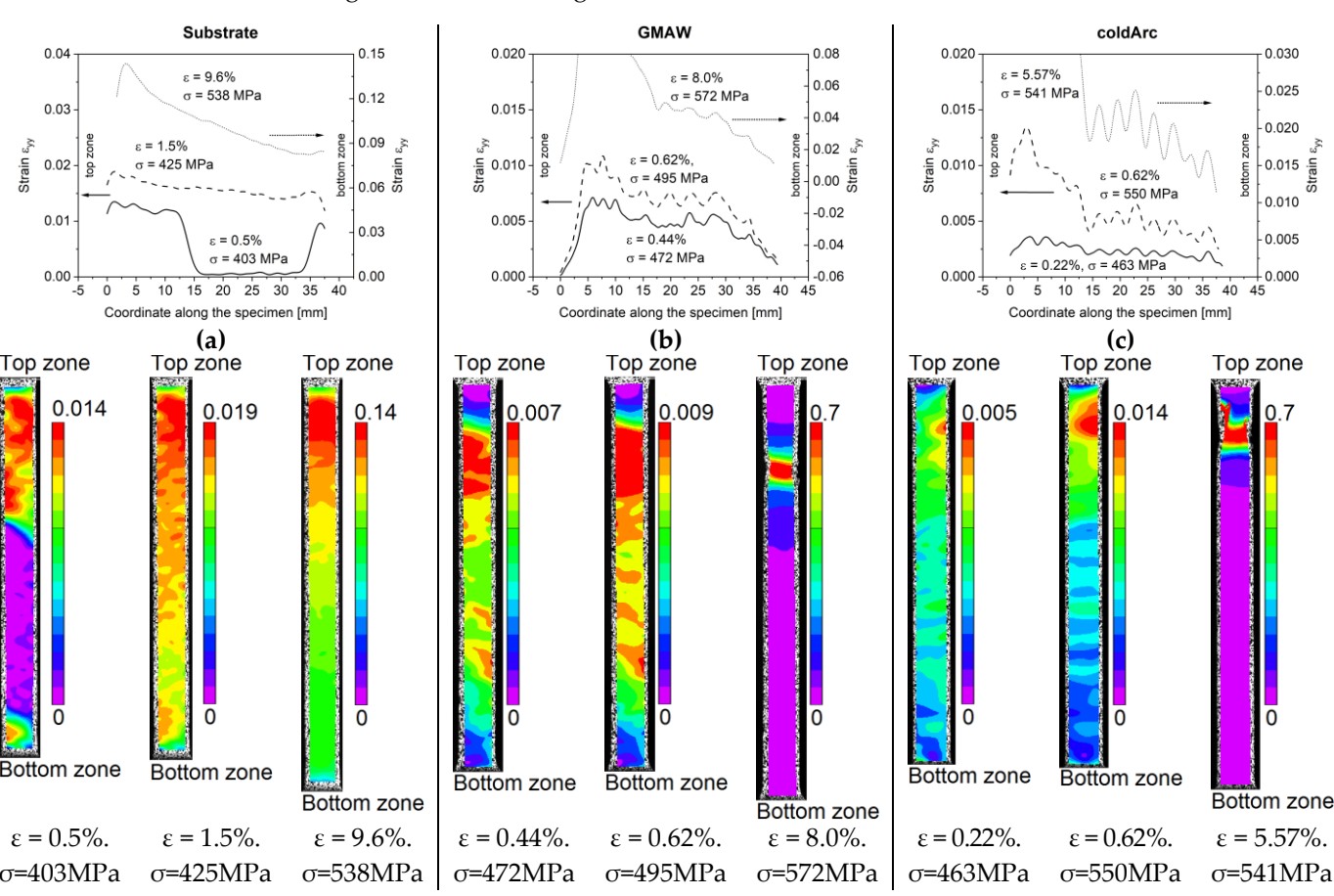

**Figure 10.** Stress–strain diagrams of tested specimens (**a**) and longitudinal strain component $\varepsilon_{yy}$ along the line in the center of the specimen cut out from the substrate (**b**), GMAW (**c**), and coldArc (**d**) walls. Evolution of strain fields obtained via digital image correlation during tensile tests of specimens cut out from the substrate (**d**–**f**), GMAW (**g**–**i**), and coldArc (**j**–**l**) walls. The width of all specimens was 4 mm.

In specimens cut from the substrate material, the uniform deformation in the elastic region continues up to a strain of 0.02% and a stress of 366 MPa, and thus, inhomogeneous deformation is observed in strain fields. Plastic behavior, according to the stress–strain diagram, occurs only at strains of 0.04%. Strain fields show an increased deformation in the upper part of the sample and also partially in the lower part. Then, the deformation front moves from the upper to the lower grip, as shown in Figure 10a ($\varepsilon$ = 0.5%) and Figure 10d—it is known as the Chernov–Luders bands [51]. Approximately at $\varepsilon$ = 1.5%. $\sigma$ = 425 MPa, the deformations along the sample have an inclined linear shape, and from this point, the formation of a neck begins (Figure 10a $\varepsilon$ = 1.5% and Figure 10e). The neck is illustrated in Figure 10a $\varepsilon$ = 9.6%; $\sigma$ = 538 MPa, and Figure 10f. The fracture is located at a distance of approximately 2.1 mm from the fillet.

Specimens cut out from the 3D-printed wall do not have a pronounced yield point. For them, at the level of 300 MPa (GMAW) and 450 MPa (coldArc), there is a slight deviation in the loading diagram, but the formation of yield plateaus.

For the specimen printed in the GMAW mode, localization of deformations begins to appear after $\varepsilon$ = 0.44% in the upper part of the specimen (Figure 10b,g). At $\varepsilon$ = 0.62%; $\sigma$ = 495 MPa, inhomogeneities are formed in the lower part of the sample in the form of bands of localized deformation (Figure 10b,h). Their formation is obviously associated with

the presence of printed layer boundaries (Figure 6), which have greater plasticity during deformation due to the higher content of the ferrite phase. The average distance between localization peaks is 4.3 mm. However, this distance exceeds the distance of torch travel in the vertical direction for each layer (~1.6 mm). Since the layers were deposited on top of each other with a slight offset, they could spread, especially when heat accumulated. Thus, the layer thickness could vary significantly from the vertical of the torch. The zone of the greatest localization of deformations before fracture is located about 7 mm from the fillet (Figure 10b,i).

For a sample printed in the coldArc mode, inhomogeneities appear in the form of stripes after $\varepsilon = 0.15\%$. Stripes are located mainly in the lower part of the sample and are associated with the layer boundaries. The heterogeneity becomes more clearly visible at $\varepsilon = 0.22\%$; $\sigma = 463$ MPa (Figure 10c,j). The average distance between localization peaks is 3.2 mm. This is more than the average vertical travel of the torch when printing a new layer (~1.8 mm) and is associated with the spreading of the layer, as was described above for GMAW.

Comparing coldArc and GMAW modes for the same strain ($\varepsilon = 0.62\%$ Figure 10c,k), it is noticeable that coldArc peaks are narrower and more frequent. When the peaks are observed in dynamics, inhomogeneities are determined more stably from frame to frame for the coldArc. They maintain the amplitude between the peak and valley, while for GMAW, the amplitude of inhomogeneities varies more frame to frame. At the end of the coldArc specimen test (Figure 10c,l), the main localization of deformations occurs at a distance of about 7.2 mm from the fillet. The coldArc mode is characterized by faster heat dissipation and higher hardness values. Hard materials are prone to strain localization, which can result in a reduction in peak spacing compared to the GMAW mode.

Stripes of local inhomogeneities associated with the layer boundaries are present in the GMAW and coldArc specimens before fracture, and residual deformations in these zones are also detected after specimen fracture (Figure 10c).

The presence of layer boundaries, as macroinhomogeneities, can significantly reduce the plasticity of specimens [12], which leads to a large difference in the strain at break of printed specimens and initial substrate steel (Table 3).

## 4. Conclusions

The structure and mechanical behavior of walls printed using multilayer arc deposition were studied. OK Autrod 13.14 wire was deposited layer-by-layer on heat-resistant steel substrate made of 12Cr1MoV. Three-dimensional printing was performed in standard GMAW mode and in coldArc mode with reduced heat input. The following conclusions were obtained:

1. As a result of the high rate of heat transfer to the substrate at the lower part of the walls, a lath martensite structure was formed both in the first layers of the walls and in the heat-affected zone in the substrate.

2. The central part of the walls consists of a mixture of acicular ferrite and allotriomorphic ferrite. Due to the accumulation of heat in the walls in this zone, the cooling rate of new layers gradually decreased, resulting in a gradual decrease in microhardness towards the top.

3. The upper part of the wall (0–10 mm from the top) is characterized by a coarse-grained structure, with distinguishable boundaries of the former austenite grains up to $290 \pm 40$ µm in size, inside which acicular ferrite was formed. The microhardness of this part is characterized by a slight increase in values as a result of rapid cooling. In the underlying layers at a distance of more than 10 mm from the top of the wall, the grain size decreases to $80 \pm 10$ µm, and interlayers of allotriomorphic ferrite are separated along the boundaries.

4. The material of the walls, compared to the substrate, has higher strength characteristics by 40–45%, and plasticity is reduced by 50–70%. The increase in strength is associated with a higher cooling rate and the formation of a fine structure of acicular ferrite. An addi-

tional factor in reducing plasticity is the layer boundaries, which have lower microhardness by 15%, observed using optical metallography and DIC.

5. The main difference between the 3D printing modes is the reduced heat input in the coldArc mode, which results in less heat accumulation and faster cooling of the wall. Thus, a more dispersed structure was formed. As a result, walls are characterized by higher values of microhardness (by 7% on average), tensile strength (by 34 MPa or 5%), and lower ductility (strain at break—6% vs. 10%).

6. The cooling rate and the level of heat input are the main factors affecting the structure formation (martensitic, bainitic, ferritic), the height and quality of the wall surface, depending on the degree of spreading of the next layer, and mechanical properties.

**Author Contributions:** Conceptualization. I.V.V. and A.I.G.; validation. I.V.V., A.I.G. and A.V.E.; investigation. A.V.E., V.M.S. and A.E.K.; writing—original draft preparation. I.V.V.; writing—review and editing. I.V.V., A.I.G. and A.V.E.; visualization. I.V.V. and A.E.K. All authors have read and agreed to the published version of the manuscript.

**Funding:** The work was performed according to the Government research assignment for ISPMS SB RAS. project No. FWRW-2021-0009 and project No. FWRW-2021-0010.

**Data Availability Statement:** Not applicable.

**Acknowledgments:** The investigations have been carried out using the equipment of Share Use Centre "Nanotech" of the ISPMS SB RAS.

**Conflicts of Interest:** The authors declare no conflict of interest.

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
