# Peer review of "Structure and Mechanical Behavior of Heat-Resistant Steel Manufactured by Multilayer Arc Deposition"

_metals, doi:10.3390/met13081375_

Round 1

Reviewer 1 Report

1.      There are still some type-errors and hard expressions. Please check it carefully.

2.      Why GMAW has thinner layers Compared to coldArc mode?

3.      How about the defined elongation rate of the static tensile tests. Why the strain is so small? How do you achieve the stress 463MPa? Why the strain in top zone is larger? A stress-strain curve is needed.

4.      Allotriomorphic ferrite and acicular ferrite should be labeled in the observation pictures. In Fig.7 a slight increase in microhardness is due to the formation of an acicular ferrite structure and a small proportion of ferrite layers. Please illustrate the observation pictures.

1.      There are still some type-errors and hard expressions. Please check it carefully.

Author Response

The authors would like to thank the Reviewer for their careful reading of the manuscript and comments which help to improve the paper. Changes in the manuscript are highlighted in yellow.

  1. There are still some type-errors and hard expressions. Please check it carefully.

The English language has been revised and corrected.

  1. Why GMAW has thinner layers Compared to coldArc mode?

The higher layer height in coldArc mode is caused by less metal spreading due to lower heat input – 0.388 kJ/mm in coldArc mode versus 0.425 kJ/mm in GMAW mode. This explains the difference in the geometric dimensions of the walls and contributes to the formation of a higher and narrower wall in the coldArc printing mode. This discussion of possible reasons for the difference in layer thicknesses has been added to the section “3.1. Wall appearance".

  1. How about the defined elongation rate of the static tensile tests. Why the strain is so small? How do you achieve the stress 463MPa? Why the strain in top zone is larger? A stress-strain curve is needed.

Static tensile tests were carried out on an Instron 5582 electromechanical machine with a crosshead travel speed of 1.5 mm/min. It is written in the manuscript in the section “Materials and methods” (line 213).

The printed walls have a structure of acicular ferrite, which is characterized by higher hardness compared to the ferrite-pearlite structure of the substrate. Therefore the plasticity (and strain at break as well) of specimens cut from the walls is lower than specimens cut from the substrate. An additional factor in reducing plasticity is the layer boundaries, which increase the strain localization.

The authors found this question unclear. The only one place where stress of 463 MPa is mentioned is a first DIC image for codlArc mode. In general mechanical stress was calculated using the following equation σ=P/A, where P – applied load, measured by load cell of testing machine and A – cross-section of the specimen at the beginning of the test.

Specimens cut from the printed walls were placed in a tensile tests the same as they were located in the printed wall. Since the lowest value of microhardness was in the upper part of the wall, the strain localization occurs in the upper part. This is visible in DIC strain fields.

Stress-strain curves were presented in the initial paper in Figure 8a, while in the revised version the diagram is shown in Figure 9. Only one curve for each type of the specimen was drawn to demonstrate general shape for each tested material.

  1. Allotriomorphic ferrite and acicular ferrite should be labeled in the observation pictures. In Fig.7 a slight increase in microhardness is due to the formation of an acicular ferrite structure and a small proportion of ferrite layers. Please illustrate the observation pictures.

The designation of the phases has been added for all metallography in Figures 4-6 of the revised version of the manuscript. To make the explanation more intelligible, the authors slightly rephrase it and note the L coordinate in brackets. It would help to find the described region in the graph of microhardness.

Reviewer 2 Report

Review report on the topic ‘Structure and mechanical behavior of heat-resistant steel man- 2 ufactured by multilayer arc deposition’. The work is presented well. The comments to improve the quality of the manuscript are listed below:

  1. Please omit the unnecessary information and add the key conclusion of the work at the end of the abstract section.
  2. Please add a separate section to discuss the novelty of the work.
  3. The introduction section is presented roughly. Add more references and try to make a bridge between current and previously published work. Add more discussion about the major problems associated with deposition and also about residual stresses: https://doi.org/10.3390/ma15207094.
  4. Add scale in each figure.
  5. How was the composition and base metal properties measured?
  6. Discuss the selction of the welding wire.
  7. Provide complete detail of the experimental setup with good quality image.
  8. Discuss the waviness obtained after deposition.
  9. Discuss the detail characterization of the interface including the laing and area map. Also discuss about the dilution and unmixed zoen formation at interface of susbtrate and deposited metal: https://doi.org/10.3390/ma14216591.
  10. Image quality is very poor and not acceptable in present form.
  11. The martensitic structure formation need more discussion.
  12. Mention the standard used for specimen preparation and also the detail about number of sample tested.
  13. Mention the stress-strain plot and tensile results in separate table along with fracture location and joint efficiency.

NA

Author Response

The authors would like to thank the Reviewer for their careful reading of the manuscript and comments which help to improve the paper. Changes in the manuscript are highlighted in yellow.

  1. Please omit the unnecessary information and add the key conclusion of the work at the end of the abstract section.

The abstract of the manuscript has been revised. The key conclusions of the work have been added to the text.

  1. Please add a separate section to discuss the novelty of the work.

A discussion on the current wire arc additive manufacturing techniques and the novelty of the present work has been added to the introduction.

  1. The introduction section is presented roughly. Add more references and try to make a bridge between current and previously published work. Add more discussion about the major problems associated with deposition and also about residual stresses: https://doi.org/10.3390/ma15207094.

An additional literature review has been performed. The current scientific directions and main issues in the investigating area have been carried out. The reference list has been extended up to 51 papers.

  1. Add scale in each figure.

Scale bars are presented in all metallographic images. For the strain fields obtained via DIC scale bars are excessive, so the width of the specimen is known and also written in the figure caption to understand the image dimensions (Figure 10).

  1. How was the composition and base metal properties measured?

The composition and properties of the base metal as well as welding wire were provided by the material supplier. These values are proved by the inspection certificate, which guarantee the properties. This comment was added to the section “2. Materials and Methods” to clarify the origin of the data in Table 1.

  1. Discuss the selection of the welding wire.

The information about the motivation for selection of welding wire and the relevance of its study has been added to the introduction.

  1. Provide complete detail of the experimental setup with good quality image.

The images in section 2 “Materials and Methods” have been rearranged and the photo of multi-axis mechanized manipulator FANUC AM-100iD has been added. The description of the technique was improved by providing detailed information about the equipment for 3D printing.

  1. Discuss the waviness obtained after deposition.

The information about the possible reasons for the formation of “wavy” layer geometry has been added in section 3.1”Wall appearance”.

  1. Discuss the detail characterization of the interface including the laing and area map. Also discuss about the dilution and unmixed zoen formation at interface of susbtrate and deposited metal: https://doi.org/10.3390/ma14216591.

To clarify the structural characterization of the printed material, the phases on metallographic images were designated.

Additionally, microhardness measurements were carried out within the boundaries of the layers for different printing modes. On average, there is a decrease in microhardness by 15% in both GMAW and coldArc mode. This result has been added to the manuscript in the section “3.3. Microhardness”

The main purpose of the work was to study the structure and mechanical properties of the walls printed at different levels of heat input. Therefore, the discussion of the “wall-substrate” interface is not the point. So the authors did not carry out sufficient results on this issue. However, the paper presents the main phases formed at the “wall-substrate” interface and describes the possible processes that occurred during their formation.

  1. Image quality is very poor and not acceptable in present form.

Uploading .docx file to the website and its subsequent conversion to .pdf format compresses the images and significantly reduce their quality. The authors provided original images to the publisher, so the final version should have good pictures.

  1. The martensitic structure formation need more discussion.

A discussion about the reasons for the formation of a martensitic structure has been added to section “3.2. metallography».

  1. Mention the standard used for specimen preparation and also the detail about number of sample tested.

The information about the number of specimen and the process for their preparation has been added in section “2. Materials and Methods». The number of tested specimens was from 3 to 6 depending on the material type. The specimen preparation was performed in several steps – emery paper with a P240 grade; P600; P1000; P2000 and finally in has been polished by diamond paste. Finally the specimen with mirror-like surface has been obtained for static testing.

  1. Mention the stress-strain plot and tensile results in separate table along with fracture location and joint efficiency.

Table 3 in the revised version of the manuscript has been rearranged. Now it additionally contains fracture location and distance between peaks of strain localization for three materials types. Moreover, to clearly illustrate the DIC analysis stress-strain plot moved close to the Table 3. After the revision Figure 10 includes strain graphs inspected by line instrument (via DIC) placed near by the DIC strain fields.

Reviewer 3 Report

The authors have analyzed the mesoscopic structure, hardness and behavior under tensile stress of two differently fabricated specimens. The samples were both produced by multilayer arc deposition, but with different heat inputs, which consequently leads to different structural and mechanical properties. Overall, the procedure and results are well presented. However, the conclusions lack a classification in the current state of research, which would enhance the manuscript enormously. I therefore recommend adding this to the manuscript. Thus, the manuscript would be publishable in Metals with minor modifications.

Other comments:

Does the OK Autrod 13.14 material contain copper as a cover material?

Can the authors provide a porosity of the fabricated materials?

In Fig. 2 b the unit [mm/mm] can be omitted.

Do the authors give an average layer height (I may have overlooked this)?

 Figure 7. it would be good to measure the hardness with higher lateral resolution. For example, in the transition areas between layers.

Table 3: Two other materials ‘Russia GOST’ are mentioned here, but they have no reference in the text. This must be added.

For Chernov-Luders band a citation is missing.

Embedding of the Conclusion in the current state of the literature.

There are some spelling errors

Author Response

The authors would like to thank the Reviewer for their careful reading of the manuscript and comments which help to improve the paper. Changes in the manuscript are highlighted in yellow.

  1. However, the conclusions lack a classification in the current state of research, which would enhance the manuscript enormously. I therefore recommend adding this to the manuscript.

An additional analysis of the literature has been carried out in the introduction. A classification of welding method and the main problems in the study area were reviewed and this information was added to the paper. The list of relevant literature has been increased.

  1. Does the OK Autrod 13.14 material contain copper as a cover material?

Copper is used as a protective coating on OK Autrod 13.14 wire. However, handbooks usually indicate the composition of the uncoated wire without copper. To evaluate the copper content, an additional analysis of the chemical composition was carried out using a portable Niton xL 3t spectrometer. On the surface of the wire, the copper content is 14%. In the process of 3D printing, a significant evaporation of copper occurs and its content does not exceed 0.3% (measurements were made on the surface of the wall). The authors believe that Cu will not play an important role in the structure formation during printing. This comments were added to the revised manuscript.

  1. Can the authors provide a porosity of the fabricated materials?

Porosity has not been measured for the printed materials, because pores are rare on SEM images. Thus it can be concluded, that the material is quite homogenous and additional porosity measurements are not obligatory.

  1. In Fig. 2 b the unit [mm/mm] can be omitted.

The units [mm/mm] in Figures 3b and 10a,b,c, have been removed

  1. Do the authors give an average layer height (I may have overlooked this)?

The layers height was mentioned in the paragraphs, where strain localization obtained via DIC was discussed. The authors considered that it also would be good to write the layers thickness in the section “3.1. Wall appearance”. “The average vertical travel of the welding torch after each layer deposition was 1.6 and 1.8 mm for the GMAW and coldArc modes, respectively.”

  1. Figure 7. it would be good to measure the hardness with higher lateral resolution. For example, in the transition areas between layers.

Measurements of microhardness along the wall height were carried out with a step of 5 mm. However, in the lower part of the wall and HAZ, where microhardness changes significantly, additional indentations were made. The step of measurements in these regions was less than 1 mm. Some of the points in Figure 8 were excluded in order to make the graph clearly visible and not overloaded with excessive data points. Microhardness measurement were also made at layers boundaries. It has been shown that microhardness in these regions is lower by 15% than in the layer. These explanations were added to the revised version of the manuscript.

  1. Table 3: Two other materials ‘Russia GOST’ are mentioned here, but they have no reference in the text. This must be added.

Russian GOST references were added in Tables 1 and 3 as well as mentioned in the text where materials are described.

  1. For Chernov-Luders band a citation is missing.

The citation for Chernov-Luders band was added in the sentence where it mentioned.

  1. Embedding of the Conclusion in the current state of the literature.

As far as literature review of the recent papers in the field of arc welding were extended and sufficiently improved, the authors prefer not to overload conclusion and leave a numerated structure to highlight the key results.

A sixth point has been added to the Conclusion, in order to highlight the key result of the article, which can contribute to the development of modern ideas about 3D printing.

  1. There are some spelling errors

The English language has been revised and corrected.